# IMPACT: Importance Weighted Asynchronous Architectures with Clipped Target Networks

**Michael Luo**
UC Berkeley
michael.luo@berkeley.edu

**Jiahao Yao**
UC Berkeley
jiahaoyao@berkeley.edu

**Richard Liaw**
UC Berkeley

**Eric Liang**
UC Berkeley

**Ion Stoica**
UC Berkeley

## Abstract

The practical usage of reinforcement learning agents is often bottlenecked by the duration of training time. To accelerate training, practitioners often turn to distributed reinforcement learning architectures to parallelize and accelerate the training process. However, modern methods for scalable reinforcement learning (RL) often tradeoff between the throughput of samples that an RL agent can learn from (sample throughput) and the quality of learning from each sample (sample efficiency). In these scalable RL architectures, as one increases sample throughput (i.e. increasing parallelization in IMPALA (Espeholt et al., 2018)), sample efficiency drops significantly. To address this, we propose a new distributed reinforcement learning algorithm, IMPACT. IMPACT extends IMPALA with three changes: a target network for stabilizing the surrogate objective, a circular buffer, and truncated importance sampling. In discrete action-space environments, we show that IMPACT attains higher reward and, simultaneously, achieves up to 30% decrease in training wall-time than that of IMPALA. For continuous control environments, IMPACT trains faster than existing scalable agents while *preserving the sample efficiency of synchronous PPO*.

## 1 Introduction

Proximal Policy Optimization (Schulman et al., 2017) is one of the most sample-efficient on-policy algorithms. However, it relies on a synchronous architecture for collecting experiences, which is closely tied to its trust region optimization objective. Other architectures such as IMPALA can achieve much higher throughputs due to the asynchronous collection of samples from workers. Yet, IMPALA suffers from reduced sample efficiency since it cannot safely take multiple SGD steps per batch as PPO can. The new agent, **Imp**ortance Weighted **A**synchronous Architectures with **C**lipped **T**arget Networks (**IMPACT**), mitigates this inherent mismatch. Not only is the algorithm highly sample efficient, it can learn quickly, training 30 percent faster than IMPALA. At the same time, we propose a novel method to stabilize agents in distributed asynchronous setups and, through our ablation studies, show how the agent can learn in both a time and sample efficient manner.

In our paper, we show that the algorithm IMPACT realizes greater gains by striking the balance between high sample throughput and sample efficiency. In our experiments, we demonstrate in the experiments that IMPACT exceeds state-of-the-art agents in training time (with same hardware) while maintaining similar sample efficiency with PPO's. The contributions of this paper are as follows:

1. We show that when collecting experiences asynchronously, introducing a target network allows for a stabilized surrogate objective and multiple SGD steps per batch (Section 3.1).

2. We show that using a circular buffer for storing asynchronously collected experiences allows for smooth trade-off between real-time performance and sample efficiency (Section 3.2).

3. We show that IMPACT, when evaluated using identical hardware and neural network models, improves both in real-time and timestep efficiency over both synchronous PPO and IMPALA (Section 4).

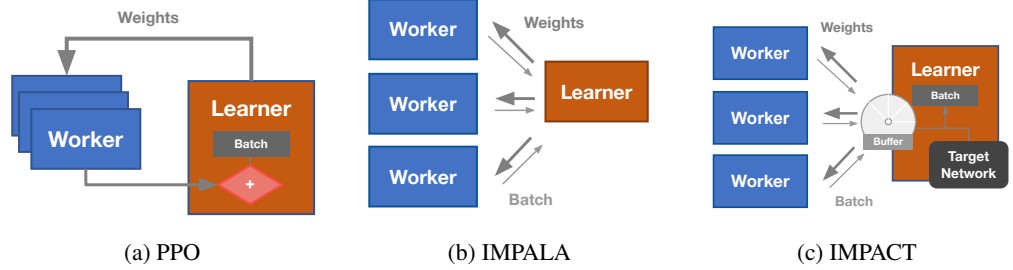

Figure 1: Architecture schemes for distributed PPO, IMPALA, and IMPACT. PPO aggregates worker batches into a large training batch and the learner performs minibatch SGD. IMPALA workers asynchronously generate data. IMPACT consists of a batch buffer that takes in worker experience and a target's evaluation on the experience. The learner samples from the buffer.

## 2 BACKGROUND

Reinforcement Learning assumes a Markov Decision Process (MDP) setup defined by the tuple $(S, A, p, \gamma, r)$ where $S$ and $A$ represent the state and action space, $\gamma \in [0, 1]$ is the discount factor, and $p : S \times A \times S \rightarrow \mathbb{R}$ and $R : S \times A \rightarrow \mathbb{R}$ are the transition dynamics and reward function that models an environment.

Let $\pi(a_t|s_t) : S \times A \rightarrow [0, 1]$ denote a stochastic policy mapping that returns an action distribution given state $s_t \in S$. Rolling out policy $\pi(a_t|s_t)$ in the environment is equivalent to sampling a trajectory $\tau \sim \mathbb{P}(\boldsymbol{\tau})$, where $\tau := (s_0, a_0, ...., a_{T-1}, s_T, a_T)$. We can compactly define state and state-action marginals of the trajectory distribution $p_\pi(s_t)$ and $p_\pi(s_t, a_t)$ induced by the policy $\pi(a_t|s_t)$. The goal for reinforcement learning aims to maximize the following objective: $J(\theta) = \mathbb{E}_{(s_t, a_t) \sim p_\pi}[\sum_{t=0}^{T} \gamma^t R(s_t, a_t)]$.

When $\theta$ parameterizes $\pi(a_t|s_t)$, the policy is updated according to the **Policy Gradient Theorem** (Sutton et al., 2000):

$$\nabla_\theta J(\theta) = \mathbb{E}_{(s_t, a_t) \sim p_\pi(\cdot)} \left[ \nabla_\theta \log \pi_\theta(a_t|s_t) \hat{A}_{\pi_\theta}(s_t, a_t) \right],$$

where $\hat{A}_{\pi_\theta}(s_t, a_t)$ is an estimator of the advantage function. The advantage estimator is usually defined as the 1-step TD error, $\hat{A}_{\pi_\theta}(s_t, a_t) = r(s_t, a_t) + \gamma \hat{V}(s_{t+1}) - \hat{V}(s_t)$, where $\hat{V}(s_t)$ is an estimation of the value function. Policy gradients, however, suffer from high variance and large update-step sizes, oftentimes leading to sudden drops in performance.

### 2.1 DISTRIBUTED PPO

Per iteration, Proximal Policy Optimization (PPO) optimizes policy $\pi_\theta$ from target $\pi_{\theta_{\mathrm{old}}}$ via the following objective function

$$L(\theta) = \mathbb{E}_{p_{\pi_{\theta_{\mathrm{old}}}}} \left[ \min \left( r_t(\theta) \hat{A}_t, \mathrm{clip} \left( r_t(\theta), 1 - \epsilon, 1 + \epsilon \right) \hat{A}_t \right) \right],$$

where $r_t(\theta) = \frac{\pi_\theta(a_t|s_t)}{\pi_{\theta_{\mathrm{old}}}(a_t|s_t)}$ and $\epsilon$ is the clipping hyperparameter. In addition, many PPO implementations use GAE-$\lambda$ as a low bias, low variance advantage estimator for $\hat{A}_t$ (Schulman et al., 2015b). PPO's surrogate objective contains the importance sampling ratio $r_t(\theta)$, which can potentially explode if $\pi_{\theta_{\mathrm{old}}}$ is too far from $\pi_\theta$. (Han & Sung, 2017). PPO's surrogate loss mitigates this with the clipping function, which ensures that the agent makes reasonable steps. Alternatively, PPO can also be seen as an adaptive trust region introduced in TRPO (Schulman et al., 2015a).

In Figure 1a, distributed PPO agents implement a synchronous data-gathering scheme. Before data collection, workers are updated to $\pi_{\mathrm{old}}$ and aggregate worker batches to training batch $D_{\mathrm{train}}$. The learner performs many mini-batch gradient steps on $D_{\mathrm{train}}$. Once the learner is done, learner weights are broadcast to all workers, who start sampling again.

## 2.2 IMPORTANCE WEIGHTED ACTOR-LEARNER ARCHITECTURES

In Figure 1b, IMPALA decouples acting and learning by having the learner threads send actions, observations, and values while the master thread computes and applies the gradients from a queue of learners experience (Espeholt et al., 2018). This maximizes GPU utilization and allows for increased sample throughput, leading to high training speeds on easier environments such as Pong. As the number of learners grows, worker policies begin to diverge from the learner policy, resulting in stale policy gradients. To correct this, the IMPALA paper utilizes V-trace to correct the distributional shift:

$$v_{s_t} = V_\phi(s_t) + \sum_{i=t}^{t+n-1} \gamma^{i-t} \left( \prod_{j=t}^{i-1} c_j \right) \rho_i (r_{i+1} + \gamma V_\phi(s_{i+1}) - V_\phi(s_i))$$

where, $V_\phi$ is the value network, $\pi_\theta$ is the policy network of the master thread, $\mu_{\theta'}$ is the policy network of the learner thread, and $c_j = \min\left(\bar{c}, \frac{\pi_\theta(a_j|s_j)}{\mu_{\theta'}(a_j|s_j)}\right)$ and $\rho_i = \min\left(\bar{\rho}, \frac{\pi_\theta(a_i|s_i)}{\mu_{\theta'}(a_i|s_i)}\right)$ are clipped IS ratios.

---

**Algorithm 1** IMPACT

---

**Input:** Batch size $M$, number of workers $W$, circular buffer size $N$, replay coefficient $K$, target update frequency $t_{\text{target}}$, weight broadcast frequency $t_{\text{frequency}}$, learning rates $\alpha$ and $\beta$
1: Randomly initialize network weights $(\theta, w)$
2: Initialize target network $(\theta', w') \leftarrow (\theta, w)$
3: Create $W$ workers and duplicate $(\theta, w)$ to each worker
4: Initialize circular buffer $C(N, K)$
5: **for** $t = 1, .., T$ **do**
6:     Obtain batch $B$ of size $M$ traversed $k$ times from $C(N, K)$
7:     If $k = 0$, evaluate $B$ on target $\theta'$, append target output to $B$
8:     Compute policy and value network gradients

$$\nabla_\theta J(\theta) = \frac{1}{M} \sum_{(i,j) \in B} \frac{\nabla_\theta \pi_\theta(s_j|a_j)}{\max(\pi_{\text{target}}(s_j|a_j), \beta \pi_{\text{worker}_i}(s_j|a_j))} \hat{A}_{V\text{-GAE}} - \eta \nabla_\theta \text{KL}(\pi_{\text{target}}, \pi_\theta)$$

$$\nabla_w L(w) = \frac{1}{M} \sum_j (V_w(s_j) - \hat{V}_{V\text{-GAE}}(s_j)) \nabla_w V_w(s_j)$$

9:     Update policy and value network weights $\theta \leftarrow \theta + \alpha_t \nabla_\theta J(\theta), w \leftarrow w - \beta_t \nabla_w L(w)$
10:    If $k = K$, discard batch $B$ from $C(N, K)$
11:    If $t \equiv 0 \pmod{t_{\text{target}}}$, update target network $(\theta', w') \leftarrow (\theta, w)$
12:    If $t \equiv 0 \pmod{t_{\text{frequency}}}$, broadcast weights to workers
13: **end for**

---

**Worker-i**

---

**Input:** Worker sample batch size $S$
1: **repeat**
2:     $B_i = \emptyset$
3:     **for** $t = 1, ..., S$ **do**
4:         Store $(s_t, a_t, r_t, s_{t+1})$ ran by $\theta_i$ in batch $B_i$
5:     **end for**
6:     Send $B_i$ to $C(N, K)$
7:     If broadcasted weights exist, set $\theta_i \leftarrow \theta$
8: **until** learner finishes

---

## 3 IMPACT ALGORITHM

Like IMPALA, IMPACT separates sampling workers from learner workers. Algorithm 1 and Figure 1c describe the main training loop and architecture of IMPACT. In the beginning, each worker copies weights from the master network. Then, each worker uses their own policy to collect trajectories

| | **PPO** | **Asynchronous PPO** | | |
|---|---|---|---|---|
| **Invariants** | $\pi_{\text{worker}} == \pi_{\text{learner}}$ | Async sampling means $\pi_{\text{worker}}$ is out of sync with $\pi_{\text{learner}}$ | | |
| **Likelihood ratio** | $\pi_{\theta}/\pi_{\text{worker}}$ | $\pi_{\theta}/\pi_{\text{worker}}$ | $\pi_{\theta}/\pi_{\text{learner}}$ | $\pi_{\theta}/\max(\pi_{\text{target}}, \beta\pi_{\text{learner}})$ |
| **Effectiveness** | In synchronous PPO, all rollouts are fully on-policy, hence $\pi_{\text{worker}}$ is the same as $\pi_{\text{learner}}$. | Since $\pi_{\text{worker}}$ may differ per worker, using this ratio results in trust region conflicts across multiple batches. | Since $\pi_{\text{learner}}$ is updated after each batch from the worker, only a single SGD step can be taken per batch. | The IMPACT objective allows for multiple SGD steps per async batch and has a stable trust region. |

Figure 2: In asynchronous PPO, there are multiple candidate policies from which the trust region can be defined: (1) $\pi_{\text{worker}_i}$, the policy of the worker process that produced the batch of experiences, (2) $\pi_{\text{learner}}$, the current policy of the learner process, and (3) $\pi_{\text{target}}$, the policy of a target network. Introducing the target network allows for both a stable trust region and multiple SGD steps per batch of experience collected asynchronously from workers, improving sample efficiency. Since workers can generate experiences asynchronously from their copy of the master policy, this also allows for good real-time efficiency.

and sends the data $(s_t, a_t, r_t)$ to the circular buffer. Simultaneously, workers also asynchronously pull policy weights from the master learner. In the meantime, the target network occasionally syncs with the master learner every $t_{target}$ iterations. The master learner then repeatedly draws experience from the circular buffer. Each sample is weighted by the importance ratio of $\frac{\pi_{\theta}}{\pi_{\text{worker}_i}}$ as well as clipped with target network ratio $\frac{\pi_{\text{worker}_i}}{\pi_{\text{target}}}$. The target network is used to provide a stable trust region (Figure 2), allowing multiple steps per batch (i.e., like PPO) even in the asynchronous setting (i.e., with the IMPALA architecture). In the next section, we describe the design of this improved objective.

### 3.1 MAXIMAL TARGET-WORKER CLIPPING

PPO gathers experience from previous iteration's policy $\pi_{\theta_{\text{old}}}$, and the current policy trains by importance sampling off-policy experience with respect to $\pi_{\theta}$. In the asynchronous setting, worker $i$'s policy, denoted as $\pi_{\text{worker}_i}$, generates experience for the policy network $\pi_{\theta}$. The probability that batch $B$ comes from worker $i$ can be parameterized as a categorical distribution $i \sim D(\alpha_1, ..., \alpha_n)$. We include this by adding an extra expectation to the importance-sampled policy gradient objective (IS-PG) (Jie & Abbeel, 2010):

$$J_{IS}(\theta) = \mathbb{E}_{i \sim D(\alpha)}\left[\mathbb{E}_{(s_t,a_t) \sim \pi_{\text{worker}_i}}\left[\frac{\pi_{\theta}}{\pi_{\text{worker}_i}}\hat{A}_t\right]\right].$$

Since each worker contains a different policy, the agent introduces a target network for stability (Figure 2). Off-policy agents such as DDPG and DQN update target networks with a moving average. For IMPACT, we periodically update the target network with the master network. However, training with importance weighted ratio $\frac{\pi_{\theta}}{\pi_{\text{target}}}$ can lead to numerical instability, as shown in Figure 3. To prevent this, we clip the importance sampling ratio from worker policy,$\pi_{\text{worker}_i}$, to target policy, $\pi_{\text{target}}$:

$$J_{AIS}(\theta) = \mathbb{E}_{i \sim D(\alpha)}\left[\mathbb{E}_{(s_t,a_t) \sim \pi_{\text{worker}_i}}\left[\min(\frac{\pi_{\text{worker}_i}}{\pi_{\text{target}}}, \rho)\frac{\pi_{\theta}}{\pi_{\text{worker}_i}}\hat{A}_t\right]\right]$$

$$= \mathbb{E}_{i \sim D(\alpha)}\left[\mathbb{E}_{(s_t,a_t) \sim \pi_{\text{worker}_i}}\left[\frac{\pi_{\theta}}{\max(\pi_{\text{target}}, \beta\pi_{\text{worker}_i})}\hat{A}_t\right]\right],$$

where $\beta = \frac{1}{\rho}$. In the experiments, we set $\rho$ as a hyperparameter with $\rho \geq 1$ and $\beta \leq 1$.

To see why clipping is necessary, when master network's action distribution changes significantly over few training iterations, worker i's policy, $\pi_{\text{worker}_i}$, samples data outside that of target policy, $\pi_{\text{target}}$, leading to large likelihood ratios, $\frac{\pi_{\text{worker}_i}}{\pi_{\text{target}}}$. The clipping function $\min(\frac{\pi_{\text{worker}_i}}{\pi_{\text{target}}}, \rho)$ pulls back

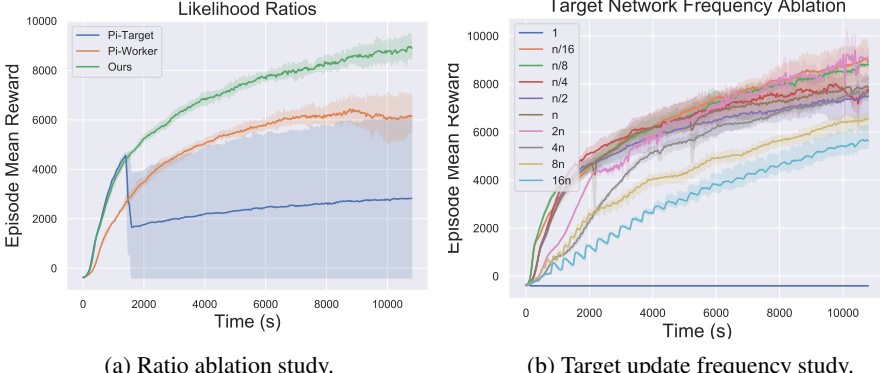

(a) Ratio ablation study.

(b) Target update frequency study.

Figure 3: Training curves of the ablation study on control benchmarks. In (a), the IMPACT objective outperforms other possible ratio choices for the surrogate loss: $R_1 = \frac{\pi_\theta}{\pi_{\text{target}}}$, $R_2 = \frac{\pi_\theta}{\pi_{\text{worker}_i}}$, $R_3 = \frac{\pi_\theta}{\max(\pi_{\text{target}}, \beta \pi_{\text{worker}_i})}$. In (b), we show the target network update frequency is robust to a range of choices. We try target network update frequency $t_{target}$ equal to the multiple (ranging from 1/16 and 16) of $n = N \cdot K$, the product of the size of circular buffer and the replay times for each batch in the buffer.

large IS ratios to $\rho$. Figure 10 in Appendix E provides additional intuition behind the target clipping objective. We show that the target network clipping is a lower bound of the IS-PG objective.

For $\rho > 1$, the clipped target ratio is larger and serves to augment advantage estimator $\hat{A}_t$. This incentivizes the agent toward good actions while avoiding bad actions. Thus, higher values of $\rho$ encourages the agent to learn faster at the cost of instability.

We use GAE-$\lambda$ with V-trace (Han & Sung, 2019). The V-trace GAE-$\lambda$ modifies the advantage function by adding clipped importance sampling terms to the summation of TD errors:

$$\hat{A}_{V\text{-GAE}} = \sum_{i=t}^{t+n-1} (\lambda\gamma)^{i-t} \left( \prod_{j=t}^{i-1} c_j \right) \delta_i V,$$

where $c_i = \min\left(\bar{c}, \frac{\pi_{\text{target}}(a_j|s_j)}{\pi_{\text{worker}_i}(a_j|s_j)}\right)$ (we use the convention $\prod_{j=t}^{t-1} c_j = 1$) and $\delta_i V$ is the importance sampled 1-step TD error introduced in V-trace.

## 3.2 Circular Buffer

IMPACT uses a circular buffer (Figure 4) to emulate the mini-batch SGD used by standard PPO. The circular buffer stores $N$ batches that can be traversed at max $K$ times. Upon being traversed $K$ times, a batch is discarded and replaced by a new worker batch.

For motivation, the circular buffer and the target network are analogous to mini-batching from $\pi_{\text{old}}$ experience in PPO. When target network's update frequency $n = NK$, the circular buffer is equivalent to distributed PPO's training batch when the learner samples $N$ minibatches for $K$ SGD iterations.

This is in contrast to standard replay buffers, such as in ACER and APE-X, where transitions $(s_t, a_t, r_t, s_{t+1})$ are either uniformly sampled or sampled based on priority, and, when the buffer is full, the oldest transitions are discarded (Wang et al., 2016; Horgan et al., 2018).

Figure 4 illustrates an empirical example where tuning $K$ can increase training sample efficiency and decrease training wall-clock time.

## 4 Evaluation

In our evaluation we seek to answer the following questions:

1. How does the target-clipping objective affect the performance of the agents compared to prior work? (Section 4.1)

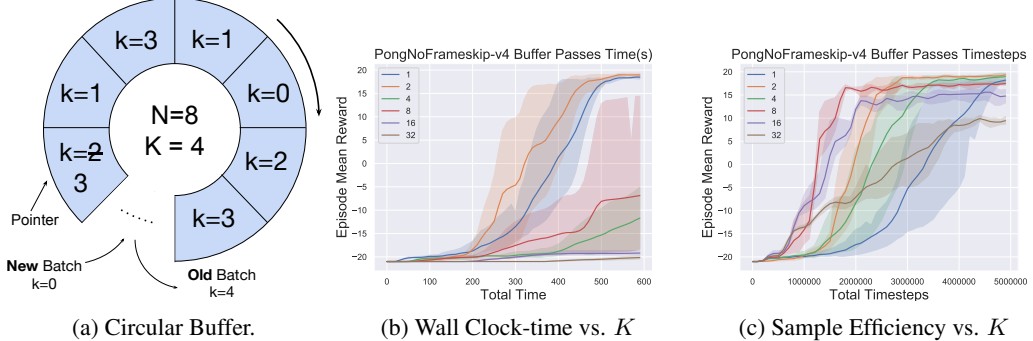

(a) Circular Buffer.  (b) Wall Clock-time vs. $K$  (c) Sample Efficiency vs. $K$

Figure 4: **(a)**: The Circular Buffer in a nutshell: $N$ and $K$ correspond to buffer size and max times a batch can be traversed. Old batches are replaced by worker-generated batches. **(b)**: The performance of IMPACT with different K in terms of time. **(c)**: The performance of IMPACT with different K in terms of timesteps. IMPACT can achieve greater timestep as well as time efficiency by manipulating $K$. $K = 2$ outperforms all other settings in time and is more sample efficient than $K = 1, 4, 16, 32$.

2. How does the IMPACT circular buffer affect sample efficiency and training wall-clock time? (Section 4.2)

3. How does IMPACT compare to PPO and IMPALA baselines in terms of sample and real-time performance? (Section 4.3)

4. How does IMPACT scale with respect to the number of workers? (Section 4.4)

## 4.1 TARGET CLIPPING PERFORMANCE

We investigate the performance of the clipped-target objective relative to prior work, which includes PPO and IS-PG based objectives. Specifically, we consider the following ratios below:

$$R_1 = \frac{\pi_\theta}{\pi_{\text{target}}} \quad R_2 = \frac{\pi_\theta}{\pi_{\text{worker}_i}} \quad R_3 = \frac{\pi_\theta}{\max(\pi_{\text{target}}, \beta \pi_{\text{worker}_i})}$$

For all three experiments, we truncate all three ratios with PPO's clipping function: $c(R) = \text{clip}(R, 1-\epsilon, 1+\epsilon)$ and train in an asynchronous setting. Figure 4(a) reveals two important takeaways: first, $R_1$ suffers from sudden drops in performance midway through training. Next, $R_2$ trains stably but does not achieve good performance.

We theorize that $R_1$ fails due to the target and worker network mismatch. During periods of training where the master learner undergoes drastic changes, worker action outputs vastly differ from the learner outputs, resulting in small action probabilities. This creates large ratios in training and destabilizes training. We hypothesize that $R_2$ fails due to different workers pushing and pulling the learner in multiple directions. The learner moves forward with the most recent worker's suggestions without developing a proper trust region, resulting in many worker's suggestions conflicting with each other.

The loss function, $R_3$ shows that clipping is necessary and can help facilitate training. By clipping the target-worker ratio, we make sure that the ratio does not explode and destabilize training. Furthermore, we prevent workers from making mutually destructive suggestions by having a target network provide singular guidance.

### 4.1.1 TARGET NETWORK UPDATE FREQUENCY

In Section 3.2, an analogy was drawn between PPO's mini-batching mechanism and the circular buffer. Our primary benchmark for target update frequency is $n = N \cdot K$, where $N$ is circular buffer size and $K$ is maximum replay coefficient. This is the case when PPO is equivalent to IMPACT.

In Figure 4(b), we test the frequency of updates with varying orders of magnitudes of $n$. In general, we find that agent performance is robust to vastly differing frequencies. However, when $n = 1 \sim 4$,

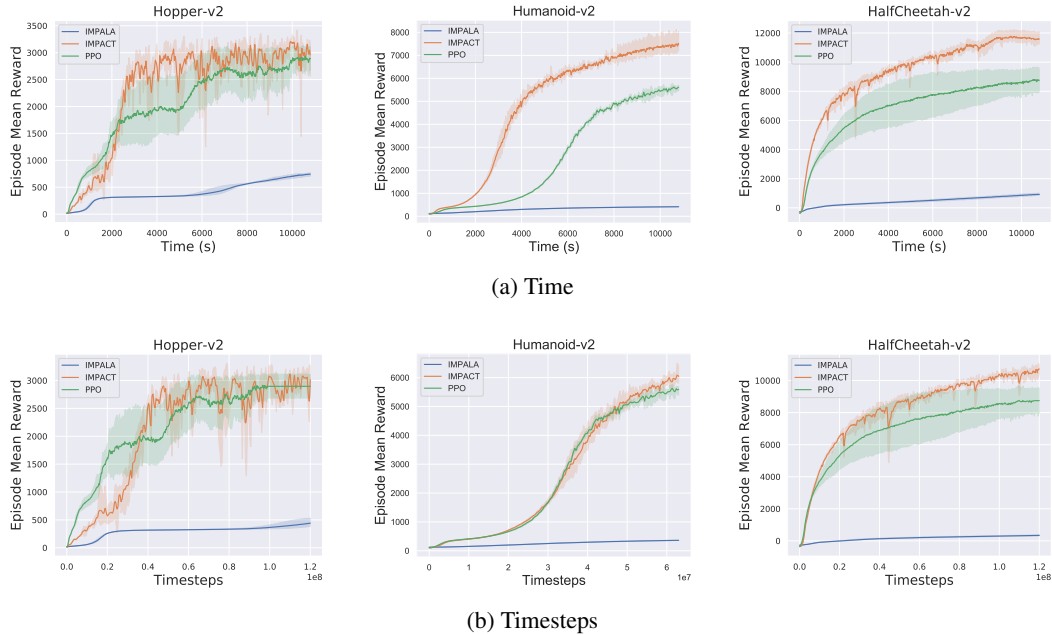

(a) Time

(b) Timesteps

Figure 5: IMPACT outperforms baselines in both sample and time efficiency for Continuous Control Domains: Hopper, Humanoid, HalfCheetah.

the agent does not learn. Based on empirical results, we theorize that the agent is able to train as long as a stable trust region can be formed. On the other hand, if update frequency is too low, the agent is stranded for many iterations in the same trust region, which impairs learning speed.

## 4.2 TIME AND SAMPLE EFFICIENCY WITH CIRCULAR BUFFER

Counter to intuition, the tradeoff between time and sample efficiency when $K$ increases is not necessarily true. In Figure 4b and 4c, we show that IMPACT realizes greater gains by striking the balance between high sample throughput and sample efficiency. When $K = 2$, IMPACT performs the best in both time and sample efficiency. Our results reveal that wall-clock time and sample efficiency can be optimized based on tuning values of $K$ in the circular buffer.

## 4.3 COMPARISON WITH BASELINES

We investigate how IMPACT attains greater performance in wall clock-time and sample efficiency compared with PPO and IMPALA across six different continuous control and discrete action tasks.

We tested the agent on three continuous environments (Figure 5): HalfCheetah, Hopper, and Humanoid on 16 CPUs and 1 GPU. The policy networks consist of two fully-connected layers of 256 units with nonlinear activation tanh. The critic network shares the same architecture as the policy network. For consistentency, same network architectures were employed across PPO, IMPALA, and IMPACT.

For the discrete environments (Figure 6), Pong, SpaceInvaders, and Breakout were chosen as common benchmarks used in popular distributed RL libraries (Caspi et al., 2017; Liang et al., 2018). Additional experiments for discrete environments are in the Appendix. These experiments were ran on 32 CPUs and 1 GPU. The policy network consists of three 4x4 and one 11x11 conv layer, with nonlinear activation ReLU. The critic network shares weights with the policy network. The input of the network is a stack of four 42x42 down-sampled images of the Atari environment. The hyper-parameters for continuous and discrete environments are listed in the Appendix B table 1 and 2 respectively.

Figures 5 and 6 show the total average return on evaluation rollouts for IMPACT, IMPALA and PPO. We train each algorithm with three different random seeds on each environment for a total time

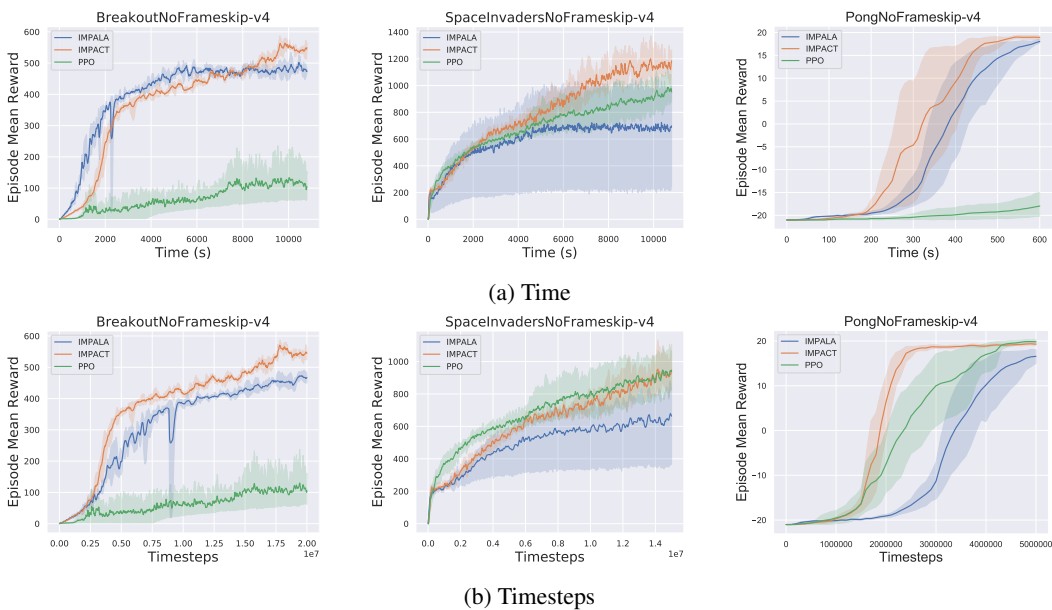

(a) Time

(b) Timesteps

Figure 6: IMPACT outperforms PPO and IMPALA in both real-time and sample efficiency for Discrete Control Domains: Breakout, SpaceInvaders, and Pong.

of three hours. According to the experiments, IMPACT is able to train much faster than PPO and IMPALA in both discrete and continuous domains, while preserving same or better sample efficiency than PPO.

Our results reveal that continuous control tasks for IMPACT are sensitive to the tuple $(N, K)$ for the circular buffer. $N = 16$ and $K = 20$ is a robust choice for continuous control. Although higher $K$ inhibits workers' sample throughput, increased sample efficiency from replaying experiences results in an overall reduction in training wall-clock time and higher reward. For discrete tasks, $N = 1$ and $K = 2$ works best. Empirically, agents learn faster from new experience than replaying old experience, showing how exploration is crucial to achieving high asymptotic performance in discrete enviornments.

## 4.4 IMPACT Scalability

Figure 7 shows how IMPACT's performance scales relative to the number of workers. More workers means increased sample throughput, which in turn increases training throughput (the rate that learner consumes batches). With the learner consuming more worker data per second, IMPACT can attain better performance in less time. However, as number of workers increases, observed increases in performance begin to decline.

## 5 Related Work

**Distributed RL architectures** are often used to accelerate training. Gorila (Nair et al., 2015) and A3C (Mnih et al., 2016) use workers to compute gradients to be sent to the learner. A2C (Mnih et al., 2016) and IMPALA (Espeholt et al., 2018) send experience tuples to the learner. Distributed replay buffers, introduced in ACER (Wang et al., 2016) and Ape-X (Horgan et al., 2018), collect worker-collected experience and define an overarching heuristic for learner batch selection. IMPACT is the first to fully incorporate the sample-efficiency benefits of PPO in an asynchronous setting.

**Surreal PPO** (Fan et al., 2018) also studies training with PPO in the asynchronous setting, but do not consider adaptation of the surrogate objective nor IS-correction. Their use of a target network for broadcasting weights to workers is also entirely different from IMPACT's. Consequently, IMPACT is able to achieve better results in both real-time and sample efficiency.

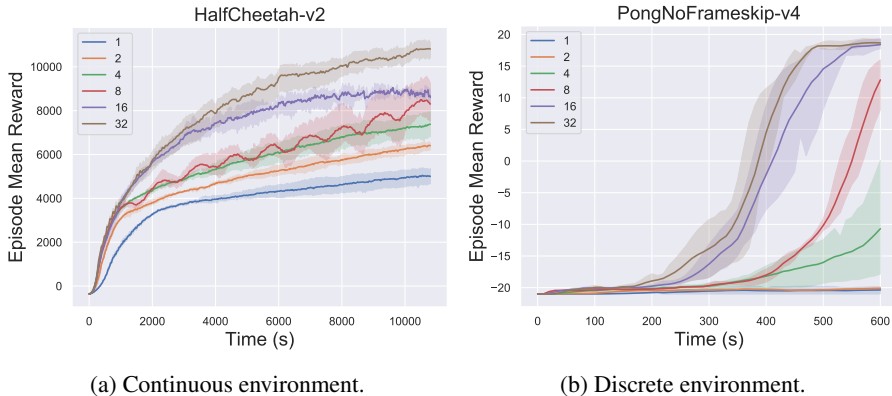

(a) Continuous environment.     (b) Discrete environment.

Figure 7: Performance of IMPACT with respect to the number of workers in both continuous and discrete control tasks

**Off-policy methods**, including DDPG and QProp, utilize target networks to stabilize learning the Q function (Lillicrap et al., 2015; Gu et al., 2016). This use of a target network is related but different from IMPACT, which uses the network to define a stable trust region for the PPO surrogate objective.

## 6 CONCLUSION

In conclusion, we introduce IMPACT, which extends PPO with a stabilized surrogate objective for asynchronous optimization, enabling greater real-time performance without sacrificing timestep efficiency. We show the importance of the IMPACT objective to stable training, and show it can outperform tuned PPO and IMPALA baselines in both real-time and timestep metrics.

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

## A    ADDITIONAL EXPERIMENTS

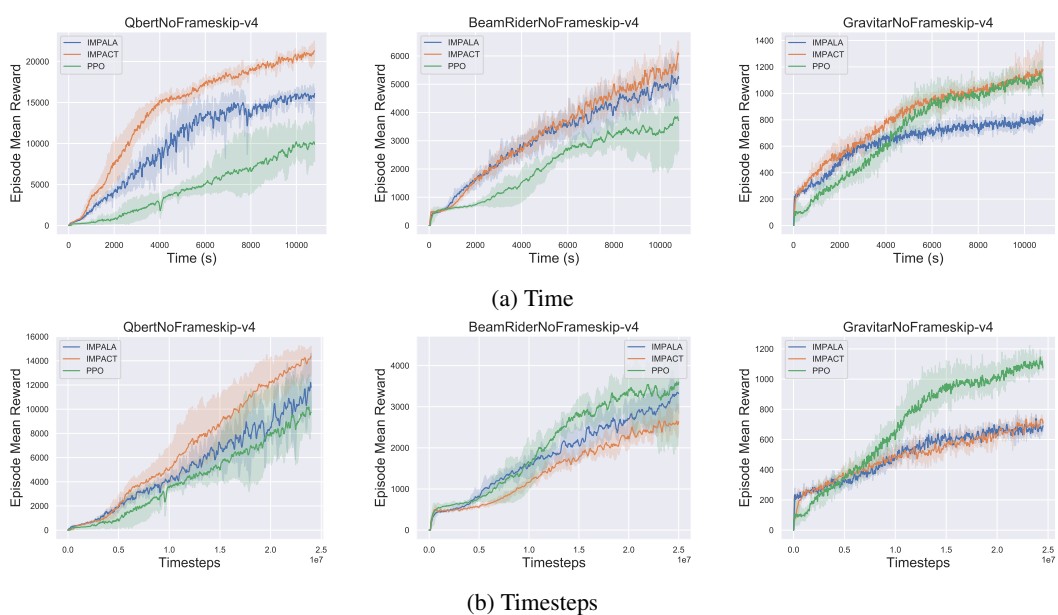

(a) Time

(b) Timesteps

Figure 8: IMPACT, PPO and IMPALA wallclock time and sample efficiency for Discrete Control Domains: Qbert, BeamRider, and Gravitar.

## B    HYPER PARAMETERS FOR ALL ENVIRONMENTS

### B.1    DISCRETE ENVIRONMENTS

| HYPERPARAMETERS | IMPACT | IMPALA | PPO |
|---|---|---|---|
| Clip Parameter | 0.3 | — | 0.1 |
| Entropy Coeff | 0.01 | 0.01 | 0.01 |
| Grad Clipping | 10.0 | 40.0 | — |
| Discount ($\gamma$) | 0.99 | 0.99 | 0.99 |
| Lambda ($\lambda$) | 0.995 | — | 0.995 |
| Learning Rate | $1.0 \cdot 10^{-4}$ | $1.0 \cdot 10^{-4}$ | $5.0 \cdot 10^{-5}$ |
| Minibatch Buffer Size (N) | 4 | — | — |
| Num SGD Iterations (K) | 2 | — | 2 |
| Sample Batch Size | 50 | 50 | 100 |
| Train Batch Size | 500 | 500 | 5000 |
| SGD Minibatch Size | — | — | 500 |
| KL Coeff | 0.0 | — | 0.5 |
| KL Target | 0.01 | — | 0.01 |
| Value Function Coeff | 1.0 | 0.5 | 1.0 |
| Target-Worker Clipping ($\rho$) | 2.0 | — | — |

Table 1: Hyperparameters for Discrete Environments.

## B.2 Continuous Environments

| Hyperparameters | IMPACT | IMPALA | PPO |
|---|---|---|---|
| Clip Parameter | 0.4 | — | 0.3 |
| Entropy Coeff | 0.0 | 0.0 | 0.0 |
| Grad Clipping | 0.5 | 0.5 | — |
| Discount ($\gamma$) | 0.995 | 0.995 | 0.99 |
| Lambda ($\lambda$) | 0.995 | — | 0.995 |
| Learning Rate | $3.0 \cdot 10^{-4}$ | $1.5 \cdot 10^{-5}$ | $3.0 \cdot 10^{-4}$ |
| Minibatch Buffer Size (N) | 16 | — | — |
| Num SGD Iterations[1](K) | 20 | — | 20 |
| Sample Batch Size | 1024 | 1024 | 1024 |
| Train Batch Size | 32768 | 32768 | 163840 |
| SGD Minibatch Size | — | — | 32768 |
| KL Coeff | 1.0 | — | 1.0 |
| KL Target | 0.04 | — | 0.01 |
| Value Function Coeff[2] | 1.0 | 0.5 | 1.0 |
| Target-Worker Clipping ($\rho$) | 2.0 | — | — |

Table 2: Hyperparameters for Continuous Control Environments

## B.3 Hyperparameter Budget

Listed below was the grid search we used for each algorithm to obtain optimal hyperparameters. Optimal values were found via grid searching on each hyperparameter separately. We found that IMPACT's optimal hyperparameter values tend to hover close to either IMPALA's or PPO's, which greatly mitigated IMPACT's budget.

### B.3.1 Discrete Environment Search

| Hyperparameters | IMPACT | IMPALA | PPO |
|---|---|---|---|
| Clip Parameter | [0.1, 0.2, 0.3] | — | [0.1, 0.2, 0.3, 0.4] |
| Grad Clipping | [10, 20, 40] | [2.5, 5, 10, 20, 40, 80] | — |
| Learning Rate ($10^{-4}$) | [0.5, 1.0, 3.0] | [0.1, 0.3, 0.5, 0.8, 1.0, 3.0, 5.0] | [0.5, 1.0, 3.0, 5.0, 8.0] |
| Minibatch Buffer Size (N) | [2,4,8, 16] | — | — |
| Num SGD Iterations (K) | [1,2,4] | — | [1,2,4,8] |
| Train Batch Size | — | — | [1000, 2500, 5000, 10000] |
| Value Function Coeff | [0.5, 1.0, 2.0] | [0.25, 0.5, 1.0, 2.0] | [0.25, 0.5, 1.0, 2.0] |
| # of Runs | 19 | 17 | 21 |

Table 3: Hyperparameter Search for Discrete Environments

---

[1]For HalfCheetah-v2, IMPACT and PPO Num SGD Iterations (K) is 32.

[2]For HalfCheetah-v2, IMPACT Value Function Coeff is 0.5.

[3]IMPALA was difficult to finetune due to unstable runs.

### B.4 CONTINUOUS ENVIRONMENT SEARCH

| HYPERPARAMETERS | IMPACT | IMPALA | PPO |
|---|---|---|---|
| Clip Parameter | [0.2, 0.3, 0.4] | — | [0.1, 0.2, 0.3, 0.4] |
| Grad Clipping | [0.5, 1.0, 5.0] | [0.1, 0.25, 0.5, 1.0, 5.0, 10.0] | — |
| Learning Rate ($10^{-4}$) | [1.0, 3.0, 5.0] | [0.1, 0.15, 0.3, 0.5, 0.8, 1.0, 3.0, 5.0] [3] | [1.0, 3.0, 5.0] |
| Minibatch Buffer Size (N) | [4,8,16] | — | — |
| Num SGD Iterations (K) | [20,26,32] | — | [20,26,32] |
| Train Batch Size | — | — | [65536, 98304, 131072, 163840] |
| KL Target | [0.01, 0.02, 0.04] | — | [0.01, 0.02, 0.04] |
| Value Function Coeff | [0.5, 1.0, 2.0] | [0.5, 1.0, 2.0] | [0.5, 1.0, 2.0] |
| # of Runs | 21 | 17 | 20 |

Table 4: Hyperparameter Search for Continuous Environments

## C  IMPALA TO IMPACT

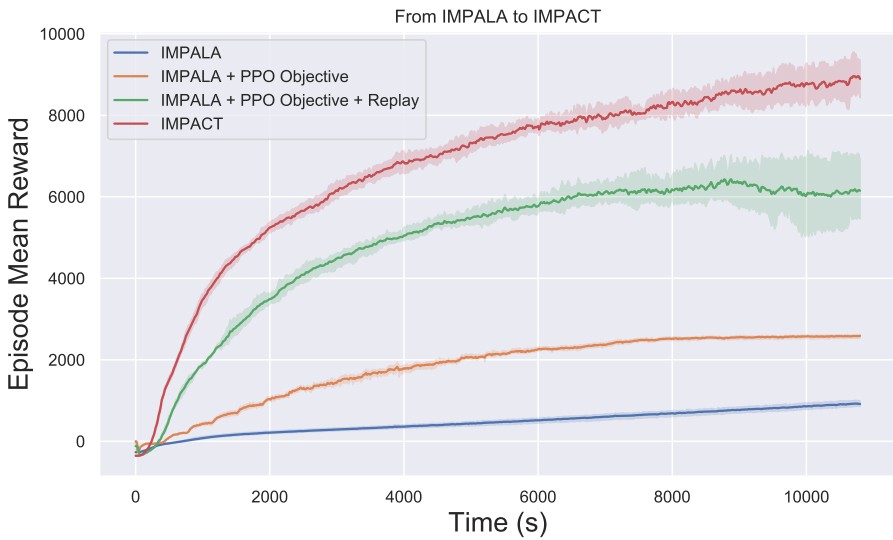

Figure 9: IMPALA to IMPACT: Incrementally Adding PPO Objective, Replay, and Target-Worker Clipping to IMPALA. The experiments are done on the HalfCheetah-v2 gym environment.

In Figure 9, we gradually add components to IMPALA until the agent is equivalent to IMPACT's. Starting from IMPALA, we gradually add PPO's objective function, circular replay buffer, and target-worker clipping. In particular, IMPALA with PPO's objective function and circular replay buffer is equivalent to an asynchronous-variant of PPO (APPO). APPO fails to perform as well as synchronous distributed PPO, since PPO is an on-policy algorithm.

## D  IMPALA IN CONTINUOUS ENVIRONMENTS

In Figure 6, IMPALA performs substantially worse than other agents in continuous environments. We postulate that IMPALA suffers from low asymptotic performance here since its objective is an importance-sampled version of the Vanilla Policy Gradient (VPG) objective, which is known to suffer from high variance and large update-step sizes. We found that for VPG, higher learning rates encourage faster learning in the beginning but performance drops to negative return later in training.

In Appendix B, for IMPALA, we heavily tuned on the learning rate, finding that small learning rates stabilize learning at the cost of low asymptotic performance. Prior work also reveals the agents that use VPG fail to attain good performance in non-trivial continuous tasks (Achiam, 2018). Our results with IMPALA reaches similar performance compared to other VPG-based algorithms. The closest neighbor to IMPALA, A3C uses workers to compute gradients from the VPG objective to send to the learner thread. A3C performs well in InvertedPendulum yet flounders in continuous environments (Tassa et al., 2018).

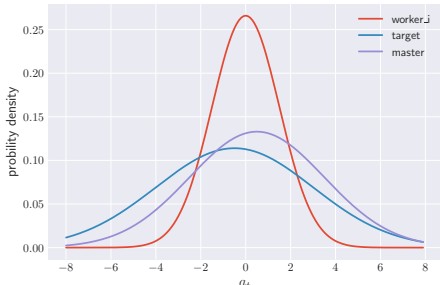 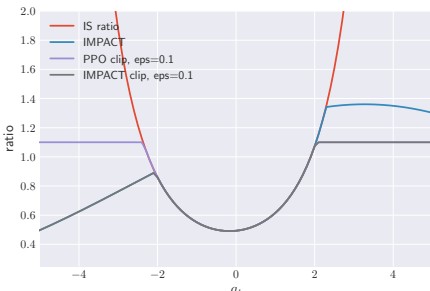

Action Distributions        Likelihood Ratios w.r.t Different Objectives

Figure 10: Likelihood ratio $r_t(\theta)$ for different objective functions, including PPO's. We assume a diagonal Gaussian policy for our policy. **Left**: Corresponding one dimensional action distributions for Worker i, Target, and Master Learner; **Right**: Ratio values graphed as a function of possible action values. IMPACT with PPO clipping is a lower bound of PPO.

## E  THE INTUITION OF THE OBJECTIVE

The following ratios represent the objective functions for different ablation studies. In the plots (Figure 10), we set the advantage function to be one, i.e. $\hat{A}_t = 1$.

- IS ratio: $\frac{\pi_\theta}{\pi_{\text{worker}_i}} \hat{A}_t$

- IMPACT target: $\min\left(\frac{\pi_{\text{worker}_i}}{\pi_{\text{target}}}, \rho\right) \frac{\pi_\theta}{\pi_{\text{worker}_i}} \hat{A}_t$

- PPO $\epsilon$-clip: $\min\left(\frac{\pi_\theta}{\pi_{\text{worker}_i}} \hat{A}_t, \text{clip}(\frac{\pi_\theta}{\pi_{\text{worker}_i}}, 1 - \epsilon, 1 + \epsilon)\hat{A}_t\right)$

- IMPACT target $\epsilon$-clip: $\min\left(\min\left(\frac{\pi_{\text{worker}_i}}{\pi_{\text{target}}}, \rho\right) \frac{\pi_\theta}{\pi_{\text{worker}_i}} \hat{A}_t, \text{clip}\left(\min\left(\frac{\pi_{\text{worker}_i}}{\pi_{\text{target}}}, \rho\right) \frac{\pi_\theta}{\pi_{\text{worker}_i}}, 1 - \epsilon, 1 + \epsilon\right) \hat{A}_t\right)$

According to Figure 10, IS ratio is large when $\pi_{\text{worker}_i}$ assigns low probability. IMPACT target $\epsilon$-clip is a lower bound of the PPO $\epsilon$-clip. In an distributed asynchronous setting, the trust region suffers from larger variance stemming from off-policy data. IMPACT target $\epsilon$-clip ratio mitigates this by encouraging conservative and reasonable policy-gradient steps.

