# OpenReview forum: "IMPACT: Importance Weighted Asynchronous Architectures with Clipped Target Networks"
_ICLR.cc/2020/Conference — Accept (Poster)_

### Official Review · AnonReviewer3 · 2019-10-23
**Official Blind Review #3**

**Rating:** 6

**Review:**

The paper studies a novel way for distributed RL training which combines the data reuse of PPO with the asynchronous updates of IMPALA. The main contribution is the observation that using a target network is necessary for achieving stable learning. I think this is an important result which seems to be validated by another ICLR submission (https://openreview.net/forum?id=SylOlp4FvH). The experimental section could definitely be improved--I was hoping to see more results on Atari or DMLab.

Two comments:
The V-trace equations (page 3 and 5) don't mention any clipping of the importance weights c and rho--can you clarify if this is a typo or if you don't use clipping?
It would be great to see experiments showing how learning curves scale with the number of workers.

-----------------------------------------------------------------------------------
Thanks for clarifying and for the extra experiments. I'm keeping my score as I still think it's appropriate.

**Experience Assessment:**

I have read many papers in this area.

**Review Assessment: Checking Correctness Of Derivations And Theory:**

N/A

**Review Assessment: Checking Correctness Of Experiments:**

I assessed the sensibility of the experiments.

**Review Assessment: Thoroughness In Paper Reading:**

I made a quick assessment of this paper.

---

> ### Author Response · Authors · 2019-11-14
> **Response to Reviewer 3**
>
> Thank you for your review!
>
> >> The experimental section could definitely be improved--I was hoping to see more results on Atari or DMLab.
>
> Additional results are in Appendix Section A; we randomly selected three additional environments from Atari-57 (Qbert, BeamRider, Gravitar).
>
> >>> It would be great to see experiments showing how learning curves scale with the number of workers.
>
> Similarly, we have attached scalability studies as an additional ablation study (Section 4.4). In short, performance increases as number of workers increases.
>
> >>> The V-trace equations (page 3 and 5) don't mention any clipping of the importance weights c and rho--can you clarify if this is a typo or if you don't use clipping.
>
> This is fixed. We used clipping in our code and there was a typo in our paper.

---

### Official Review · AnonReviewer2 · 2019-10-23
**Official Blind Review #2**

**Rating:** 6

**Review:**

This paper introduces IMPACT which is a distributed RL algorithm that shortens training time of RL systems while maintaining/ improving the sample efficiency. It is built on top of the famous PPO algorithm (https://arxiv.org/abs/1707.06347). The authors break down the novel component of their model into three categories: target network, circular buffer, and importance sampling. They evaluate the effectiveness of each component through different experiments.

Overall the paper is well-written and the ideas are communicated clearly. I like how the evaluation is done in different environments (discrete and continuous action-space) and improves the results independent of the task settings.

One question: you mentioned in section 4.3: "For fairness, same network hyperparameters were used across PPO, IMPALA, and IMPACT." I suppose it would be a fair comparison if you choose the hyperparameters for each algorithm separately ( according to the highest value they achieve on the measured metric.) How did you end up choosing the hyperparameters for your own experiments? Are they fine-tuned for IMPACT?

It seems like IMPACT is not always doing better than PPO in the discrete control domain as shown in Figure 6. Specifically, in part (a) it looks like PPO is beating both IMPALA and IMPACT for BreakoutNoFrameskip and PongNoFrameskip. It would be nice if authors could do an analysis of these cases and add a discussion as to why this is happening.

Overall I think this is an interesting paper which can motivate more work in this area.


------------------------------------------------------------------------------------------------------------------------------------------------
Updates:
I would like to thank the authors for their response. I have read the revised version and it looks good to me.


**Experience Assessment:**

I do not know much about this area.

**Review Assessment: Checking Correctness Of Derivations And Theory:**

I assessed the sensibility of the derivations and theory.

**Review Assessment: Checking Correctness Of Experiments:**

I did not assess the experiments.

**Review Assessment: Thoroughness In Paper Reading:**

I read the paper at least twice and used my best judgement in assessing the paper.

---

> ### Author Response · Authors · 2019-11-14
> **Response to Reviewer 2**
>
> >>> One question: you mentioned in section 4.3: "For fairness, same network hyperparameters were used across PPO, IMPALA, and IMPACT." I suppose it would be a fair comparison if you choose the hyperparameters for each algorithm separately ( according to the highest value they achieve on the measured metric.)
>
> For network hyperparameters, we meant that we used the same policy architecture across all agents. We edited the paper to more clear about this, thanks for pointing this out! We did do a sweep across other hyperparameters to select the optimal one for each algorithm. In Appendix B, we show the hyperparameter search space used and the final hyperparameters chosen.
>
> >>>How did you end up choosing the hyperparameters for your own experiments? Are they fine-tuned for IMPACT?
>
> We performed coordinate descent on the search spaces show in Appendix B, searching across several choices of learning rate, batch size, gradient clipping, etc. for each algorithm.
>
> >>> It seems like IMPACT is not always doing better than PPO in the discrete control domain as shown in Figure 6. Specifically, in part (a) it looks like PPO is beating both IMPALA and IMPACT for BreakoutNoFrameskip and PongNoFrameskip. It would be nice if authors could do an analysis of these cases and add a discussion as to why this is happening.
>
> In Figure 6, IMPACT (orange) beats both IMPALA (Blue) and PPO (Green) in terms of time. Breakout is close, where IMPALA is first beating IMPACT, but IMPACT eventually attains higher performance. The bottom charts show that IMPACT has comparable timestep-efficiency performance relative to PPO.

---

### Official Review · AnonReviewer1 · 2019-10-23
**Official Blind Review #1**

**Rating:** 3

**Review:**

The paper proposes a new distributed algorithm for reinforcement learning. The paper lists three main contributions: a target network for stabilizing the surrogate objective, a circular buffer, and truncated importance sampling.

I'm not that familiar with RL, however I'm very familiar with distributed training in other contexts. Therefore, the significance of the contributions in the RL domain is a bit unclear to me. However, the contributions in the area of distributed training is relatively fair. The introduction of a circular buffer is not very novel. Further, the trade-offs / adaption of update frequency etc. are standard ways to improve performance in distributed training.

The evaluation of the proposed algorithm is reasonably well done (considering the page limits), with a suitable set of benchmarks (although relatively few). The results are promising and could have a significance for practitioners.

**Experience Assessment:**

I have read many papers in this area.

**Review Assessment: Checking Correctness Of Derivations And Theory:**

I assessed the sensibility of the derivations and theory.

**Review Assessment: Checking Correctness Of Experiments:**

I assessed the sensibility of the experiments.

**Review Assessment: Thoroughness In Paper Reading:**

I read the paper at least twice and used my best judgement in assessing the paper.

---

> ### Author Response · Authors · 2019-11-14
> **Response to Reviewer 1**
>
> Thank you for your response! We believe our results are significant for the following reasons:
>
> 1. Our method is the first policy-gradient based agent to perform well across both continuous control and discrete environments, in both real-time and wall-time performance. Our results show improvements over IMPALA in continuous control tasks and PPO in real-time efficiency for discrete tasks.
>
> 2. While IMPACT does incorporate well known techniques such as replay to improve its performance, we also introduce novel algorithmic improvements such as the stabilized surrogate objective, which our ablation studies show are critical for the best performance. These algorithmic innovations are critical for "allowing" the more standard techniques such as replay and asynchrony to be used effectively.
>
> In addition, we also address your specific comments on the paper:
>
> >> Further, the trade-offs / adaption of update frequency etc. are standard ways to improve performance in distributed training.
>
> We studied the tradeoffs between N and K for circular buffer and update frequency for the target network as ablation studies, not a way to improve performance. The techniques we used to improve performance are what you listed in the beginning of the review: target network, circular buffer, and IS-clipping. To make the role of these components more clear, in Appendix C we introduce a further study interpolating between the IMPALA and IMPACT algorithm, showing clear improvements for each component added.

---

### Official Review · AnonReviewer4 · 2019-11-04
**Official Blind Review #4**

**Rating:** 6

**Review:**

Reinforcement learning (RL) training speed is broadly evaluated on two dimensions:  sample efficiency (the number of environment interactions required) and wall-clock time.  Improved wall-clock training time has been achieved through distributed actors and learners, but often at the expense of sample efficiency.  IMPACT repurposes successful concepts from deep RL - the target network, importance sampling and a replay buffer to demonstrate improvements on both axes in on three continuous environments and three games from the Atari Learning Environment.

Positives
This was a well-written paper proposing to address the sample efficiency of distributed RL algorithms.  The diagrams of the algorithm were also well-done.  Improving the sample efficiency of algorithms is an important objective and the approaches followed here are sensible.

Additionally, the ablations and examination of the sensitive hyperparameters of the algorithm are useful analyses.  These indicate relative insensitivity to the target network update frequency, but both the importance sampling equation and the circular buffer hyperparameters are described.


Negatives
IMPACT introduces additional hyperparameters which are tuned for each continuous control task and discrete control task.  However, there is no description of the hyperparameter tuning budget allocated to IMPACT, PPO, IMPALA.

Regarding the discrete environment, the game selection should be elaborated upon and if sufficient compute is available, the algorithm should be tested elsewhere.  Specifically, it is atypical (though not necessarily incorrect) to tune specific hyperparameters for each game in the Atari Learning Environment.  Traditionally, algorithms have been justified as robust and useful by the lack of need to tune per game.  Table 4 demonstrates a high degree of tuning for IMPACT due to game-specific changes for clip param, grad clip, lambda, num sgd iter, train_batch_size, value function loss coeff, kl coeff.  However, Table 5 (IMPALA) and Table 6 (PPO) have fewer noted changes.

Small nits:
- Define advantage in the policy gradient equation
- Figure 4 is ahead of Figure 3 in the compiled LaTeX


Questions
- How were the discrete control games selected?
- What was the hyperparameter tuning budget for IMPACT versus PPO or IMPALA?
- If a fixed hyperparameter budget is allocated in advance and new environments are randomly selected, does IMPACT favorably compare to IMPALA and PPO?
- IMPALA performs remarkably badly in the three continuous control tasks, even on wall-clock time.  What validations have been done here to ensure the algorithm is operating as intended?

I will increase my rating if the robustness and improvements of this algorithm can be validated in randomly chosen games/continuous control environments for a fixed hyperparameter budget for IMPACT and both baselines.  Also, the IMPALA baseline should be validated for the continuous control tasks - it's surprising that this once SOTA-algorithm flounders in even simple tasks like Hopper-v2 or HalfCheetah-v2.


-----------------
Update:
The authors have addressed my initial concerns carefully through extra experiments and details in the Appendix and I have updated my rating accordingly.  Thanks!


**Experience Assessment:**

I have published one or two papers in this area.

**Review Assessment: Checking Correctness Of Derivations And Theory:**

I assessed the sensibility of the derivations and theory.

**Review Assessment: Checking Correctness Of Experiments:**

I carefully checked the experiments.

**Review Assessment: Thoroughness In Paper Reading:**

I read the paper at least twice and used my best judgement in assessing the paper.

---

> ### Author Response · Authors · 2019-11-14
> **Response to Reviewer 4**
>
> Thank you for this detailed review! We found your suggestions very helpful and have implemented all your suggestions in our revision.
>
> One note: the IMPALA curves for the requested new experiments are still WIP in Appendix A. We will update them again shortly once runs complete (hopefully in a few hours).
>
> >> How were the discrete control games selected?
>
> Our three original discrete control games were chosen based on popular environments in existing distributed Reinforcement libraries such as Intel Coach. Based on your suggestions, we also selected three more games: two from Mnih et al. (2016), and also Gravitar, and have added results in Appendix A. All these games were selected by us without knowing their performance on any particular algorithm beforehand.
>
> >> What was the hyperparameter tuning budget for IMPACT versus PPO or IMPALA?
>
> The budget per algorithm was similar: (19, 17, 21) and (21, 17, 20) distinct trials for (IMPACT, IMPALA, and PPO) respectively on the discrete and continuous environments. We show the search space, which we searched over with coordinate descent, in Appendix B.3 and B.4. We note that the optimal hyperparameters for IMPACT are quite close to IMPALA's and PPO's.
>
> >> If a fixed hyperparameter budget is allocated in advance and new environments are randomly selected, does IMPACT favorably compare to IMPALA and PPO?
>
> Yes. We tested our final hyperparameter choice on Qbert, BeamRider, and Gravitar. In terms of real-time efficiency, IMPALA was the best for all three new environments. Timestep-wise, the agent did well on Qbert and BeamRider, but was beaten by PPO on Gravitar.
>
> >>I will increase my rating if the robustness and improvements of this algorithm can be validated in randomly chosen games/continuous control environments for a fixed hyperparameter budget for IMPACT and both baselines.
>
> We have chosen three additional discrete environments at random (Qbert, BeamRider, Gravitar) from the Atari-57 and we included their performance in Appendix A. Furthermore, we have evaluated our environments with a fixed set of hyperparameters for all of these environments: the same as chosen for previous experiments.
>
> We found IMPACT to attain very similar gains in performance when we used this universal hyperparameter baseline. The exact hyperparameters used are shown in Appendix B.1 and B.2.
>
> >>The IMPALA baseline should be validated for the continuous control tasks - it's surprising that this once SOTA-algorithm flounders in even simple tasks like Hopper-v2 or HalfCheetah-v2.
>
> We investigated this further, and believe our baseline is reasonable. We support this as follows:
>
> In Appendix C we added an ablation study interpolating between the IMPALA and IMPACT algorithms. This was done by incrementally adding the PPO objective, replay, and target network respectively on top of the baseline IMPALA implementation. It shows that each IMPACT component provides an improvement on top of IMPALA. We also note that, beyond these component changes, the code of our IMPACT implementation and IMPALA baseline are identical.
>
> In Appendix D, we discuss prior work that support our result that VPG agents fail to attain good performance in non-trivial continuous tasks (Achiam, 2018). Our results with IMPALA reaches similar performance compared to other VPG-based algorithms.

---

> > ### Comment · AnonReviewer4 · 2019-11-14
> > **Updated Rating**
> >
> > Thanks for your careful analysis of IMPACT and the additional details in the Appendix.  I have updated my rating accordingly.

---

### Decision · Program_Chairs · 2019-12-19

**Decision:**

Accept (Poster)

**Comment:**

The authors propose a novel distributed reinforcement learning algorithm that includes 3 new components: a target network for the policy for stability, a circular buffer, and truncated importance sampling. The authors demonstrate that this improves performance while decreasing wall clock training time.

Initially, reviewers were concerned about the fairness of hyper parameter tuning, the baseline implementation of algorithms, and the limited set of experiments done on the Atari games. After the author response, reviewers were satisfied with all 3 of those issues.

I may have missed it, but I did not see that code was being released with this paper. I think it would greatly increase the impact of the paper at the authors release source code, so I strongly encourage them to do so.

Generally, all the reviewers were in consensus that this is an interesting paper and I recommend acceptance.